# It Is Not Just About Storing Energy: The Multifaceted Role of Creatine Metabolism on Cancer Biology and Immunology

**DOI:** 10.3390/ijms252413273

**Published:** 2024-12-11

**Authors:** Yuheng Geng, Susan L. DeLay, Xiaoyang Chen, Jason Miska

**Affiliations:** 1Department of Neurological Surgery, Feinberg School of Medicine, Northwestern University, 676 N St. Clair, Suite 2210, Chicago, IL 60611, USA; 2Malnati Brain Tumor Institute of the Lurie Comprehensive Cancer Center, Feinberg School of Medicine, Northwestern University, Chicago, IL 60611, USA

**Keywords:** creatine, tumor metabolism, brain tumor, immunology

## Abstract

Creatine, a naturally occurring compound in mammals, is crucial in energy metabolism, particularly within muscle and brain tissues. While creatine metabolism in cancer has been studied for several decades, emerging studies are beginning to clarify the sometimes-contradictory role creatine has in either the promotion or inhibition of cancer. On one hand, creatine can directly enhance anti-tumor CD8+ T-cell activity and induce tumor apoptosis, contributing to antitumor immunity. Conversely, other studies have shown that creatine can facilitate cancer cell growth and migration by providing an energy source and activating several signaling pathways. This review will examine what is known about creatine in cancer biology, with a focus on understanding its roles across different cellular compartments. Lastly, we discuss the emerging roles of creatine metabolism, providing exciting new insights into this often-overlooked pathway. This review highlights the complex role of creatine in cancer development and treatment, offering insights into its potential as both a therapeutic target and a risk factor in oncogenesis.

## 1. Introduction

Creatine is an organic compound that occurs naturally in mammals and serves various purposes in the musculoskeletal and central nervous systems. It is typically stored in skeletal muscle as free creatine (~40%) or phosphocreatine (~60%). Guanidinoacetic acid (GAA), a creatine precursor, is synthesized in the kidney by guanidinoacetate methyltransferase (GATM) using glycine and L-arginine. In the next step, GAA is transported to the liver, where it is methylated by N-guanidinoacetate methyltransferase (GAMT) to form creatine [1]. Afterward, creatine can be circulated throughout the body by the membrane transporter SLC6A8 (also called the Creatine Transporter, CrT) [2] (Figure 1A). Creatine can enter the body in two ways: through the diet or via arginine-derived synthesis. Red meat, poultry, and fish are potent dietary sources of creatine. Arginine, which is synthesized in the urea cycle in the kidney, is a precursor of GAA, which is required for the biosynthesis of creatine. Unlike GAA, which is transported to the brain through the blood–CSF barrier (BCSFB) with the help of the taurine transporter TauT (SLC6A6), creatine enters the brain through the blood–brain barrier (BBB) via SLC6A8, also termed the creatine transporter (CrT) [3]. The CrT gene is located on the X chromosome and operates as a plasma membrane protein, facilitating the movement of creatine into and out of cells in an ion-based process [4]. This transporter is expressed by microcapillary endothelial cells (MCECs) at the BBB but is not detected in surrounding astrocytes.

There are two traditional roles of creatine in biological systems. The first is to ‘store’ excess phosphate through the action of creatine kinase (CK), which phosphorylates creatine to produce phosphocreatine (PCr). This excess phosphate can be stored and used under metabolic stress, such as muscle contraction. The second is termed the creatine phosphagen system, which plays a crucial role by facilitating the transfer of mitochondrial ATP to the cytoplasm, supporting various cellular functions. This process ensures an efficient energy supply for activities requiring rapid ATP utilization (Figure 1B). Four types of CK have been identified. CKMT1 and CKMT2 are mitochondrial isoforms that primarily catalyze the forward reaction, converting creatine to PCr. CK brain type (CKB) and CK muscle type (CKM) are cytoplasmic isoforms that primarily catalyze the reverse reaction, converting PCr back to creatine while generating ATP. Although these reactions are reversible, the cellular energy state often determines their direction. This system serves as a critical energy buffer, facilitating the interconversion of ADP and ATP and storing phosphate during metabolic stress [5]. For example, CK and PCr are key in converting ADP to ATP for muscle contraction and energy storage. Muscle store PCr, which CK can break down to resynthesize ATP when needed [6]. While the highest concentrations of creatine and the highest specific CK activity are found in skeletal muscle, the compound is responsible for far more actions in other areas of the body.

One of the most studied systems in which creatine exerts essential effects is within the central nervous system (CNS). Indeed, patients with creatine transporter deficiency exhibit severe developmental delays, mental retardation, and seizures alongside a muscle weakness phenotype [7]. Therefore, creatine transport is essential to brain health. Indeed, several studies have shown that, on a broad level, creatine promotes cognitive performance [8,9,10], delays Huntington’s Disease progression [11] (although this is controversial [12]), and prevents traumatic brain injury [13] and several other pathologies [14]. Despite creatine’s obvious importance to CNS health, the specific pathway for promoting cognitive health still needs further examination. A few studies have examined this. For example, one study showed that creatine promotes the differentiation of striatal neuronal precursor cells and reduces neuronal cell loss [15]. Another study hypothesized that creatine could be a neurotransmitter [16]. Another mechanism suggests that it can prevent excitotoxic damage to neurons from malonate lesions and 3-nitropropionic acid neurotoxicity [17].

Creatine also participates in fat metabolism, while beige fat cells utilize the creatine-driven cycles to support energy generation during stimulation like a cold environment or ADP depletion [18]. This metabolism fuels a “futile cycle”, generating heat as a byproduct of the reaction [19]. Furthermore, CK activity maintains membrane potentials and calcium homeostasis and restores ion gradients. Additionally, creatine and CK inhibit the mitochondrial permeability transition, which results in a higher rate of ATP turnover in gray matter areas of the brain [1].

Creatine also plays a crucial role in preventing cell death in other tissues; for instance, in tumor necrosis factor (TNF)-induced apoptosis, creatine inhibits cell death. In a study comparing hepatic injury in wild-type and liver-specific creatine kinase (CK)-overexpressing transgenic mice, the results demonstrated that both liver CK expression and dietary creatine supplementation effectively prevented liver cell apoptosis [20]. This discovery revealed the anti-apoptotic effects of creatine occur via maintaining mitochondrial functionality under stress [20]. Creatine has also been shown to prevent death from hypoxia/reoxygenation injury [21]. Therefore, creatine can play several pro-survival roles in cells under stress.

The role of creatine in tumor biology, however, is less clear. Several seemingly contradictory observations about this pathway in cancer therapy have been made. Simply put, some studies have shown that creatine helps prevent cancer, while others suggest that it has a promoting effect on cancer. Much of this may be attributed to the tools and technologies available at the time, as more recent insights have begun to clarify the role of creatine metabolism in tumor biology. Below, we will discuss these studies to clarify these observations.

## 2. Creatine Prevents Tumor Growth

While some studies suggest that creatine can directly inhibit tumor growth, few studies have periodically highlighted this effect. An initial study in 1999 examined how mice responded to systemic administration of creatine or cyclocreatine during tumor growth. This study demonstrated significant growth inhibition of LS174T human colon adenocarcinomas grown subcutaneously in immunodeficient nude mice when treated with either creatine or the cell-permeable analog cyclocreatine [22]. The authors proposed that creatine kinase phosphorylates creatine and enhances the buildup of substrates within cells, consequently amplifying their indirect antitumor effect [22]. In another more recent study, rats with Walker-256 breast cancer were treated with creatine, and while overall animal survival did not change, tumor growth was modestly decreased [23]. In another recent study, authors demonstrated that creatine could synergize with methylglyoxal to kill cancer cells directly [24].

Apart from a few studies demonstrating creatine having direct anti-tumor effects, recent evidence suggests that creatine primarily exerts its anti-tumor effects by enhancing anti-tumor immunity, particularly through its impact on host immunity. In a pioneering study, researchers found that tumor-infiltrating T-cells significantly upregulate SLC6A8 (CrT), and global deficiency of CrT in CD8 T-cells prevented anti-tumor immune responses [25]. Furthermore, they found that exogenous creatine supplementation enhanced the effectiveness of anti-PD1 therapy in a mouse melanoma model. These results illustrate that T-cells use creatine as an energy source when defending against cancer cells. 

Indeed, recent research has shown that creatine is crucial for CD8+ T-cell functionality and survival. Models of conditional knockout of SLC6A8 and CKB with the T-cell compartment reveal that creatine is essential in maintaining CD8+ T-cell homeostasis and expansion during CMV infection [26]. Further experiments identified that CKB activity is essential for driving signaling linked to mTORC1 when T-cells are activated [26]. These findings suggest that CKB might be located near mTORC1 on lysosomes to aid in the effective phosphorylation of mTORC1 protein targets [26]. This is likely achieved by maintaining high ratios of ATP to ADP in the local cellular environment. In addition to CKB and the creatine transporter gene, glycine amidino transferase (GATM) may also play a role in T-cell creatine production as GATM is the rate-limiting step in creatine production [27]. However, no evidence exists that T-cells can produce their creatine.

In another recent study, authors discovered that creatine could also enhance anti-tumor responses driven by macrophages. In this study, systemic creatine supplementation enforced an “M1-like” phenotype in macrophages to promote anti-tumor immunity mediated by supporting anti-tumor CD8+ T-cell responses [28]. It is unclear to what extent this occurs across tumors, as an influential study identified creatine as being essential to driving the M2-like phenotype of macrophages [29]. In this study, authors found that exogenous creatine could suppress STAT1 signaling while promoting IL-4/STAT6 reprogramming through the SWI/SNF complex. This has been supported by subsequent studies showing that STAT6 upregulates de novo creatine biosynthesis in macrophages [30]. However, it is essential to appreciate that the M1/M2 dichotomy is not as simple as previously thought [31]. While the pro-reparative and pro-inflammatory phenotypes exist in vivo, the clear delineation of the two phenotypes is insufficient to recapitulate the heterogeneity seen in vivo [31]. Furthermore, the pro- or anti-tumor effects of M1-like and M2-like cells are still debated in cancer. In some models, M1-like macrophages can promote tumorigenesis [32]; in others, they can inhibit tumor growth [31]. Therefore, understanding the exact role of creatine biology in macrophages regarding cancer is challenging to understand from a binary role. Indeed, recent work from our laboratory, which we will discuss below, found that metabolism in macrophages can be used to fuel brain tumor growth [33]. This fits with the more established view of creatine metabolism in cancer; it is a pro-malignant process, which we discuss in detail below.

## 3. Creatine Promotes Malignancy

Indeed, there is robust historical evidence for elevated creatine kinase activity in cancer. In studies conducted as early as the 1970s, elevated levels of Creatine Kinase B could be detected in the plasma of patients with prostate cancer [34], gastric cancer [35], lung adenocarcinoma [36], and breast cancer [37]. While these historical studies support the idea that creatine metabolism is elevated in cancer, therapeutics to target this pathway have only recently been used. The first studies used the compound cyclocreatine, an analog of creatine that can be transferred to PCr by creatine kinases [38]. Earlier research revealed its toxic effects on both rat and chicken embryos [39]. However, unlike phosphocreatine, which can be easily transferred to create ATP, the formation of PCr makes it difficult for it to be dephosphorylated by creatine kinase. Thus, this compound can be used to compete with normal creatine directly and, as a result, deplete the phosphocreatine gradient inside cancer cells. In glioblastoma, increased CKB expression causes substantial production of PCr, which inhibits the polyubiquitination of the chromatin regulator BRD2. Disruption of PCr by cyclocreatine degrades BRD2, thereby inhibiting chromosome separation and cancer cell proliferation [40]. Studies in nude mice with colon adenocarcinoma and prostate cancer models revealed specific anticancer effects of cyclocreatine by comparing the growth and migration of cancer cells between treatment groups [22,41]. Administering cyclocreatine supplements hindered the advancement of cancer in models of prostate cancer lacking the PTEN and SPRY2 genes, as well as in a liver metastasis model using xenografts [41]. Together, these findings illustrate the antitumor effect of cyclocreatine and provide promising research guidelines for developing future cancer treatments.

After creatine is transported into the cell through SLC6A8 or directly synthesized through GATM/GAMT [42], it can react with intracellular signals and promote cancer growth. One of these pathways has been discovered in orthotopic colorectal cancer and breast cancer models. The supply of creatine triggers the MPS1 signaling pathway axis, which upregulates SMAD2/3 and Snail/Slug [43]. MPS1, known as Monopolar Spindle 1, is a kinase that participates in chromosome–spindle attachment and cell cycle checkpoints [44]. Snail and Slug play important roles in cancer metastasis by promoting epithelial-mesenchymal transition (EMT) to improve cancer cell mobility. Unlike usual colorectal cancer (CRC) metastasis caused by elevated TGF-β signaling, creatine-mediated MPS1 signaling can bypass TGF-β receptors and directly activate the TGF-beta pathway to increase Snail/Slug [45,46]. As a result, creatine supplementation enhances the metastasis of the tested cancer model [45] (Figure 2—top panel). Since creatine is so important in cancer, its transporter could be a promising target for cancer treatment. Indeed, the SLC6A8 inhibitor RGX-202-01 successfully induced colon cancer apoptosis by controlling creatine and phosphocreatine levels [2]. Previous studies have shown that SLC6A8 inhibition prevents the uptake of phosphocreatine/creatine produced from the extracellular space CKB during cancer cell migration [47]. As a result, both exogenous and endogenous creatine sources are blocked by SLC6A8 inhibition. SLC6A8 helps cancer cells collect energy via import and protects cancer cells from reactive oxygen species (ROS). The accumulation of creatine through SLC6A8 results in triple-negative breast cancer (TNBC) cell redox homeostasis under a hypoxic environment [48]. At the mechanistic level, within cells, creatine enhances the defense against antioxidants by decreasing the activity of mitochondria and the rates at which oxygen is consumed [48]. This reduction in oxygen consumption helps to decrease the buildup of harmful reactive oxygen molecules within the cell. As a result, this process activates a signaling pathway involving AKT-ERK, which in turn safeguards the survival of hypoxic TNBC cells [48] (Figure 2—bottom panel). Instead of working independently, SLC6A8 also works with the acyl-CoA synthetase (Acsbg1). This phenomenon was detected in a study of obesity-accelerated tumor progression [49]. Under obesity conditions, the adipocyte-specific creatine biosynthesis pathway complements the upregulation of Acsbg1 with the help of SLC6A8 for intercellular creatine uptake [49]. These findings revealed the importance of the tumor microenvironment for the tumor-promoting efficiency of creatine.

As a group of enzymes that directly cause creatine and phosphocreatine transformation and undoubtedly play important roles in cancer promotion, two CKs are believed to be closely connected with oncology: CKMT1 in mitochondria and CKB in the brain. For example, a recent study found that low serum levels of CKB predict poor survival in pancreatic cancer [50]. Another study analyzed CKMT1 expression levels in more than thirty types of tumors and revealed an increase in CKMT1 expression during cancer growth [51]. Researchers have also identified a strong correlation between creatine kinase expression and cancer prognosis through GEPIA2 and Kaplan–Meier plots [51]. CKMT plays a significant role in cancer development and metastasis, especially when it is phosphorylated. For example, in Her2+ breast cancer, CKMT1 is phosphorylated and stabilized to support cancer proliferation [52] (Figure 2—middle panel); specifically, this CKMT phosphorylation is mediated by the ABL oncogene, which phosphorylates CK at the Y153 site [52]. The phosphorylation of CKMT prevents its degradation and provides a stable ATP source for cancer cell proliferation. Fortunately, this mutation site is expressed only in Her2+ breast cancer [52], which provides a promising target for further drug treatment.

**Figure 2 ijms-25-13273-f002:**
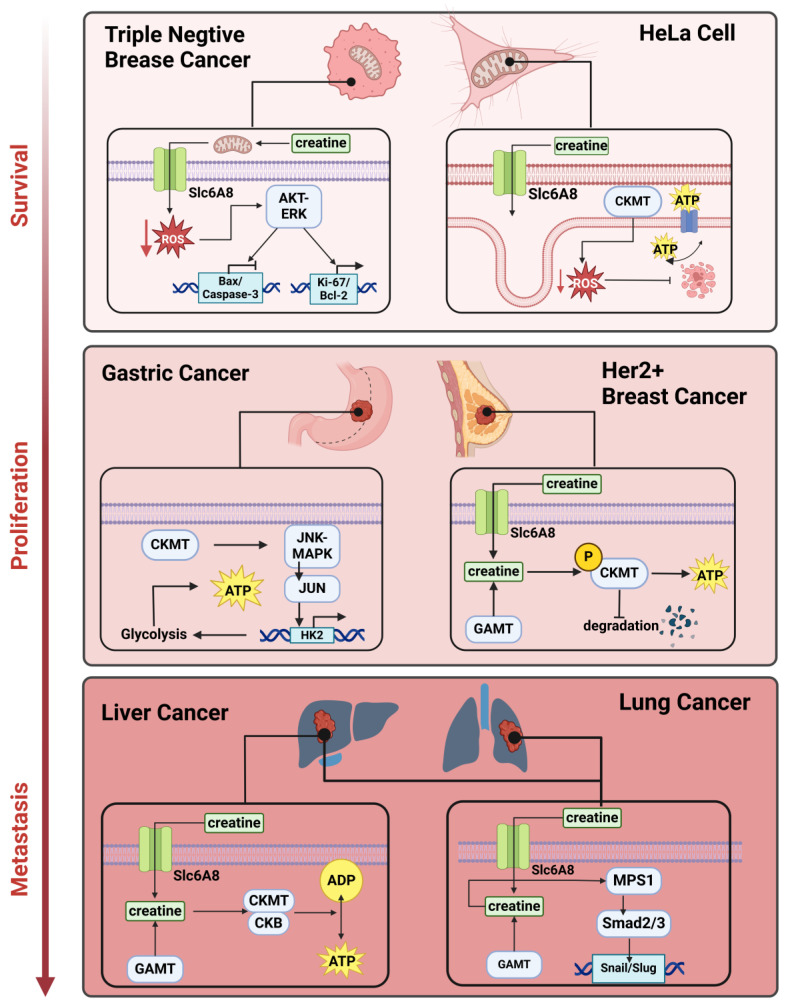
The role of creatine in the promotion of cancer malignancy. Creatine promotes various cancers through different mechanisms. These pathways include ROS manipulation, ATP supplementation, and signaling pathway activation. In HeLa cells, CKMT helps stabilize ATP levels and charges inside mitochondria, restricts ROS production, and stabilizes PT pores to prevent apoptosis [53]. Stabilizing CKMT through phosphorylation prevents its degradation and supports breast cancer growth [52]. Creatine’s ability to activate pathways such as MPS1, AKT, and MAPK supports lung, breast, and gastric cancer growth and migration [43,48,54]. The original energy-storing and boosting process also plays a key role in supporting the growth of cancers such as liver cancer and colon cancer [42,55]. In the context of pancreatic cancer, Tian et al. demonstrated that CARD9 deficiency enhances pancreatic cancer growth by impairing dendritic cell maturation through SLC6A8-mediated creatine transport. This finding suggests that targeting creatine transport and immune modulation could offer new therapeutic strategies for improving immune responses and limiting tumor growth in pancreatic cancer [56].

In human gastric cancer, CKMT can enhance the JNK-MAPK/JUN signaling pathway, which controls cell proliferation and apoptosis [54]. This signaling pathway then upregulates HK2-dependent glycolysis, of which HK2 is the rate-limiting enzyme that determines the speed of glycolysis. CKMT can also combine with the mitochondrial membrane to stabilize the PT–pore system and regulate cell apoptosis [53]. Knocking down CKMT in cancer cells strongly induces cell caspase-dependent apoptosis and mitochondrial membrane voltage changes [53]. During this process, CKMT might help stabilize the ATP/ADP ratio and protect mitochondria from excessive ROS production to prevent apoptosis. Thus, CKMT promotes cancer proliferation and metastasis, both directly and indirectly (Figure 2). The creatine kinase brain type (CKB) also plays a crucial role in regulating cell cycle progression by managing ATP levels at sites with high energy demands and influencing essential signaling pathways for cell division and cytoskeleton reorganization. This ATP buffering system, known as the phosphocreatine (PCr)-CK shuttle or PCr-CK circuit, is prominent in vertebrates [57]. It compensates for energetic signaling defects caused by adenylate kinase (AK) knockdown. It collaborates with nucleoside diphosphate kinase (NDPK) and the AMPK pathway to sense and fuel ATP/GTP-dependent processes [58]. Additionally, CKB promotes the production of mitochondrial ATP but suppresses glycolytic ATP production. This serves to prevent the activation of mitochondrial permeability transition pore (mPTP) to streamline ATP generation in mitochondria [59]. In muscle cells, CKB is recruited by sarcomeres, which maintain ATP levels during muscle contraction [60]. CKB is also recruited by various proteins, such as myosin, actin thin filaments, and titin, to facilitate rapid ATP regeneration [60]. This regulation is essential for cell cycle progression from the G1 and G2 to the S and M phases. CKB affects checkpoints from the G2 to the M phase by influencing the energy reservoirs needed for mitosis and initiating mitotic signaling pathways [60]. CKB localizes to the mitotic spindle and chromosomes during cell division, and its presence is critical for cell viability, particularly under metabolic stress conditions such as hypoxia or hypoglycemia. Overall, the role of CK in managing local ATP levels is vital for cell motility and cytoskeleton rearrangement, suggesting its potential involvement in cancer cell survival and regulation of metastasis.

## 4. Emerging Concepts About Creatine Biology in Cancer

Although many studies have shown that creatine is closely associated with cancer, there is much we have yet to study. One of the most unclear aspects is how other creatine metabolites contribute to malignancy. For example, γ-Aminobutyric acid transporter 2 (GAT2/SLC6A13) aids in the uptake of guanidinoacetate (GAA), a starting material for synthesizing Cr. No study has identified the role of GAA transport in tumors. This is especially important in the context of GBM biology. Several emerging studies have shown that GAA is the most abundantly changed metabolite isolated from formalin-fixed paraffin-embedded (FFPE) GBM tissue [61] and tumor interstitial fluid compared to normal tissue [62]. These highlight that creatine metabolites may exert unique biology in brain tumors. Another putative creatine efflux transporter has been recently described in proximal tubule cells of the kidney (MCT12) [63]. The role of this transporter and creatine biology has never been studied beyond this publication and may also be relevant for tumors. In summary, there are unique transporters and creatine metabolites crucial for creatine biology [64], yet their role in cancer is almost entirely unknown. Another emerging concept is that creatine transport supports dendritic cell maturation and promotes anti-tumor immunity against pancreatic cancer [56].

Other than the transport of creatine metabolites, which is little studied, the source of these metabolites also needs to be examined. Indeed, there is high production of creatine through the liver and kidney, but it is less clear in cancer. For example, in our recent study, we found that de novo creatine biosynthesis was being performed by myeloid cells in GBM [33]. In this study, we found that under hypoxic conditions, tumor cells upregulate SLC6A8, while simultaneously, myeloid cells increase creatine production, supplying it to the tumor cells (Figure 3).

**Figure 3 ijms-25-13273-f003:**
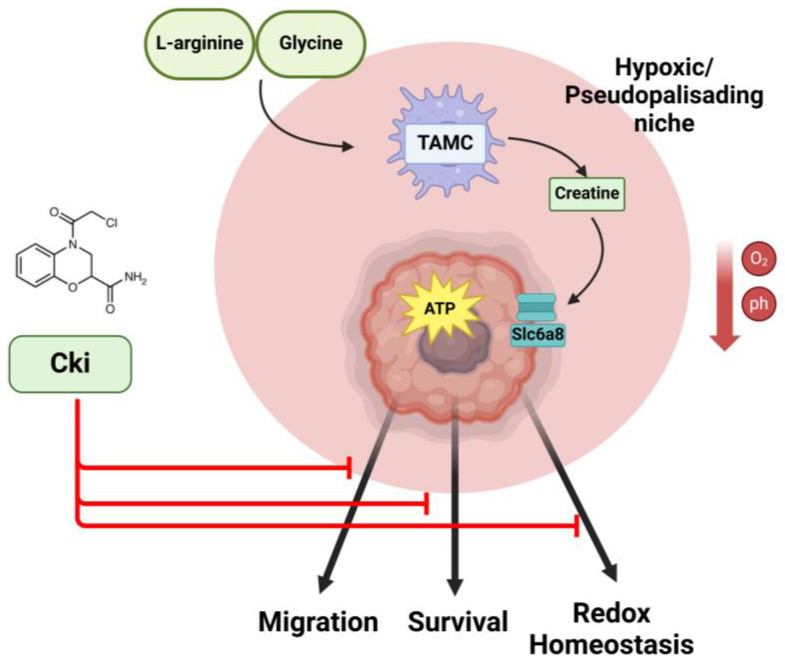
Creatine metabolism is essential for glioblastoma cell survival and can be targeted by newly developed therapeutics. In the hypoxic and acidic microenvironment of glioblastoma, tumor-associated myeloid cells (TAMCs) are recruited and synthesize creatine, which is subsequently delivered to glioblastoma cells that upregulate the creatine transporter, Slc6a8, under metabolic stress [33]. Newly developed inhibitors of creatine kinase (Cki) can potently inhibit GBM cell migration, promote redox stress, and sensitize GBM cells to other redox-perturbing compounds [65].

These findings reveal a new creatine utilization pathway in GBM in which myeloid cells are recruited to produce creatine for growth and migration. In addition to its impact on cancer cells, creatine influences the functionality of macrophages by controlling macrophage polarization. The polarization of macrophages serves as an important signal for activating macrophages to exhibit different phenotypes [66]. By inhibiting the IFN-γ signaling pathway and promoting the IL-4 pathway, the transport of creatine through SLC6A8 maintains the functional balance of M1 and M2 macrophages [29]. These findings demonstrate the importance of creatine in controlling macrophage immunosuppressive functions and may be a promising topic for future research.

In another recent study, authors found that matrix stiffness controlled the expression of CKB, and a stiffer matrix could induce creatine metabolism and enhance metastasis of pancreatic tumors through the activation of YAP signaling [67]. Tumor suppressor gene p53 has also been found to arrest CK production by upregulating GAMT and can participate in fatty acid oxidation (FAO) to regulate ATP production [68]. Thus, the mechanoresponsive element of creatine metabolism demonstrates a level of regulation far beyond simple substrate availability. 

Recently, a new creatine kinase inhibitor was developed to inhibit CKMT and CKB in cancer treatment. This compound (CKi) targets a specific active cysteine site on all creatine kinases, covalently bonding to and inhibiting creatine kinases [69]. The effectiveness of this drug was tested in AML cell lines, and the results revealed that the compound was on-target, reduced phosphocreatine levels, and induced significant cell death [69]. The principle of this drug is to bond with CK to form a covalent copolymer to prevent ATP from binding with the enzyme. In a recent study by our group, we found that CKi could potently inhibit the migration and invasion of glioblastoma (GBM) cells [65] (Figure 3). Mechanistically, we found that CKi induced significant redox stress, and the addition of cell-permeable glutathione could rescue the migratory properties of these cells. Lastly, we found that Cki could be combined with another redox inhibitor, Buthionine sulfoximine (BSO) or ferroptosis inducer (RSL3), to promote GBM cell death. This is concordant with another recent study that demonstrated that IGF1 enhances CKB binding with GPX4 and phosphorylates GPX4 to prevent its degradation by HSC70 [70]. As GPX4 is one of the main regulators of a cell process termed ferroptosis [71], this study demonstrates that creatine-mediated phosphorylation may directly prevent redox stress/ferroptosis in cells. The results suggest that the creatine phosphagen system could phosphorylate other targets to change their behavior. These recent developments highlight how much more there is to learn about creatine biology in cancer.

## 5. Future Approaches and Conclusions

In the past, we only understood creatine as an important energy support for muscle health [72]. However, as more mechanisms of creatine have been discovered, the whole field has become more complex. Since creatine could also support energy for cancer migration and metabolism, consuming supplementary creatine for body health should be viewed more conservatively. Fortunately, there is currently no direct evidence to suggest that creatine induces tumorigenesis. Still, studies have suggested that creatine may act as a precursor to HCA and increase the risk of cancer [73].

In cancer-related treatment, creatine is gradually becoming a novel and promising target for future metabolic therapy. Two possible approaches could be focused on developing new treatment technologies. The first is based on the signaling pathway involved in creatine metabolism. As mentioned earlier, creatine is expected to serve as a novel target to help target and inhibit cancer-promoting pathways, such as the AKT, MPS1, and MAPK pathways, to achieve the goal of inhibiting cancer cell growth [43,48,53]. The second approach involves energy and metabolic pathways based on creatine and phosphocreatine. Blockade of these pathways inhibits cancer growth or even induces apoptosis [65]. Inhibiting the growth and migration of cancer cells is relatively easy to understand, as inhibiting creatine eliminates one of the important energy supply pathways for cancer cells. However, its role in the induction of cancer cell death still requires further research.

Although creatine has been increasingly demonstrated to promote cancer, we cannot forget its benefits. Creatine not only provides energy for T-cells but also plays a crucial role in neural development [74]. Therefore, accurate targeting is crucial when creatine is used to address cancer. Some promising methods may include lipid nanoparticles, self-microemulsifying drug delivery, and in situ gel drug delivery [75]. As we gain a deeper understanding of creatine, the functions of this metabolic system are gradually being revealed. Creatine is crucial in energy homeostasis and participates in regulating signals in the body. However, we have not yet successfully integrated these fragmented findings, so more experimental research is needed in the future to map the metabolic function of creatine and identify the decisive influencing factors.

## 6. Methods

To explore the role of creatine in glioblastoma (GBM), we reviewed our own data and noted the lack of a comprehensive resource for new researchers in this field. We first compiled relevant papers we had read and developed a preliminary structure for the manuscript, including an Introduction, Sections on the positive and negative roles of creatine in cancer, and a Discussion. Next, we conducted literature searches in PubMed and Google Scholar using keywords such as “creatine”, “phosphocreatine”, and “creatine kinase” in combination with terms like “cancer” and “T-cell immunology”. Review articles were examined to gain a broader understanding of creatine and to identify knowledge gaps. From there, we focused on original research articles by following citations from review papers and targeted searches. This approach allowed us to build a comprehensive foundation for our manuscript.

## Figures and Tables

**Figure 1 ijms-25-13273-f001:**
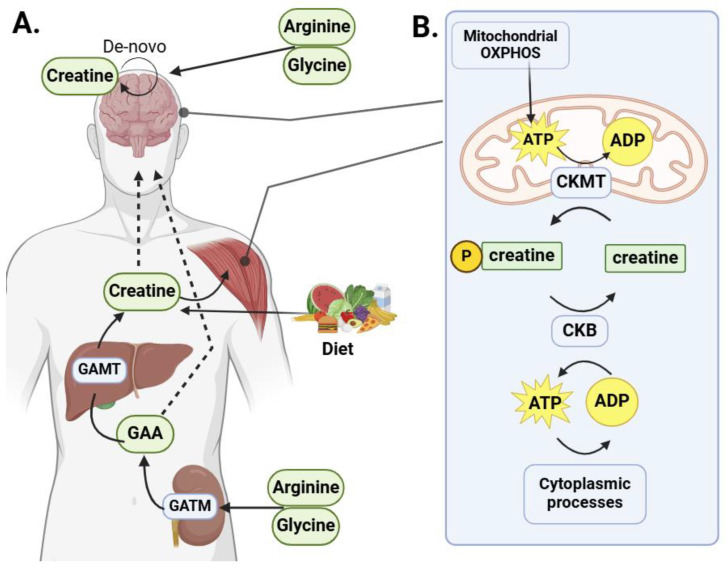
Simple schematic of creatine metabolism. In (**A**), glycine amidinotransferase (GATM) transfers L-arginine and glycine from the kidney to form guanidinoacetic acid (GAA). Further chemical synthesis in the liver, aided by guanidinoacetate methyltransferase (GAMT), produces creatine. Creatine is then transported throughout the body, primarily consumed by muscle tissue and, to some extent, the brain, via the SLC6A8 transporter. While there is evidence of de novo creatine synthesis in the brain, the relative contributions of systemic versus CNS production remain unclear. GAA can also cross the blood–brain barrier into the CNS. In (**B**), the creatine phosphagen system facilitates the transfer of mitochondrial ATP to the cytoplasm for various cellular functions. Note: All creatine metabolic enzyme reactions are reversible.

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
