# Peer review of "It Is Not Just About Storing Energy: The Multifaceted Role of Creatine Metabolism on Cancer Biology and Immunology"

_ijms, 2024, doi:10.3390/ijms252413273_

Round 1

Reviewer 1 Report

Comments and Suggestions for Authors

This manuscript is about the role of creatine metabolism in cancer biology and immune response.

The topic is of interest, but the presentation and flow of the manuscript are not good. 

Several concerns should be addressed

Introduction

Lines 26-29, please rephrase to me clearer.

"Guanidinoacetic acid (GAA), a creatine precursor, is synthesized in the kidney by glycine amidinotransferase,  mitochondrial (GATM), and then transported to the liver, where it is methylated by N-guanidinoacetate methyltransferase (GAMT) to form creatine" It is not clear the sequence of the events described  and what is GATM in comparison with GAA.

 Lines 29-30

"Afterward, the transporter SLC6A8 (also called the Creatine Transporter: CrT) can transport it across the membrane and" I would say that a transporter acts as a molecular complex that can transport something else. This sentence is repetitive.

Line 41.42. "Therefore, this suggests that the BBB is semipermeable to creatine." why this statement? Please explain better.

Lines 48-50 "While the highest concentrations of creatine and the highest specific CK activity are found in skeletal muscle, the compound is responsible for far more actions in other areas of the body." Please be more specific, which are these actions? And where do they take place?

Lines 54-56. The assertion that "creatine promotes cognitive performance [8-10]" is somehow in contrast to what is stated in lines 58-59: "Its specific role in promoting cognitive health is unclear"

Line 62. What is: "it can prevent excitotoxic damage to neurons" There is no explanation for this effect. Please be clearer.

Lines 63-68, This portion on the futile cycles in metabolism, is not contextualize.d with the title of this paper.

Lines 70-76. The authors talked about creatine in the liver. However, the English language is not good. Also, the content of the paper cited is not well explained. Indeed, this paper concludes: "The results indicate that TNFalpha-induced apoptosis was inhibited in CK transgenic mice livers by maintaining mitochondrial function" but the creatine was used in a diet of mice in a specific and artificial model compared to normal diet with less creatine.

The relevance of creatine could be on the good state of mitochondria, and this is not indicated.

Lines 77-81. This part is not specific. The authors claimed that there are contradictory results, but they did not mention any of specific information to give at least an idea on what the authors are talking about.

Chapter 2

Lines 83-135. In this part of the paper, the authors reported, in a quite confusing manner, the role of creatine as an antitumor factor. This part is a list of assertions, not well organized and presented. 

It appears that creatine can affect CD8 T cells and macrophages, besides having some direct inhibiting effect on tumor cells. Unfortunately, the authors do not make this clear distinction starting the chapter, but they insert confusing assertions. It is difficult to follow,

The statement that M1 and M2 paradigm is an artifact found in vitro is really too strong. I certainly agree that it is not possible to distinguish well between M1 and M2. However, the literature is plenty of papers claiming the difference between these two cell populations and the possibility to distinguish several M2 subsets, at least in murine models. In addition, there are several clinical trials based on this distinction trying to reconvert M2 into M1. Almost all the papers regarding the role of creatine come from the murine models (in some instances, quite difficult to reproduce and artificially built).

Anyway, the flow of information is not schematic and some portions are difficult to read.

For instance, the assertion:"This fits in with the more established view of creatine in cancer; it’s a promalignant process, which we discuss in detail below" I do not consider creatine a "process". Also, it is somehow obvious that creatine (a battery for T cells) can favor tumor growth, as a tumor needs ATP. 

Chapter 3

As for the chapter 2, the chapter 3 is the description of the protumor role for creatine. 

The MPS1 pathway is not described but mentioned, this reviewer can have an idea of this pathway, but all readers know any kind of biochemical pathway. The authors have to introduce the pathway and the link among the other pathways to give the right flow of information.

Near the end of the chapter, the authors introduced the CK, but these enzymes (and the types of these enzymes) have not been considered in detail in the introduction. This makes it difficult to link the different portions of the chapter.

In the last chapter, the authors reported some of the data derived from relevant publications of the authors on GBM. I would suggest inserting a figure or an additional panel into figure 2. Indeed, the observations mentioned are of relevance to GBM biology.

The font in the figures is small and difficult to see.

Comments on the Quality of English Language

I am not an expert on the English language but I think several corrections should be made.

Author Response

Reviewer #1

Several concerns should be addressed

Introduction

  1. Lines 26-29, please rephrase to me clearer.

"Guanidinoacetic acid (GAA), a creatine precursor, is synthesized in the kidney by glycine amidinotransferase,  mitochondrial (GATM), and then transported to the liver, where it is methylated by N-guanidinoacetate methyltransferase (GAMT) to form creatine" It is not clear the sequence of the events described  and what is GATM in comparison with GAA.

1R: Here is the rephrased version:

Guanidinoacetic acid (GAA), a creatine precursor, is synthesized in the kidney by guanidinoacetate methyltransferase (GATM) using glycine and L-arginine. Next step, GAA transported to the liver, where it is methylated by N-guanidinoacetate methyltransferase (GAMT) to form creatine.

  1.  Lines 29-30

"Afterward, the transporter SLC6A8 (also called the Creatine Transporter: CrT) can transport it across the membrane and" I would say that a transporter acts as a molecular complex that can transport something else. This sentence is repetitive.

2R: We rephrased the original sentences to make if more fluent:

Afterward, creatine can be circulated throughout the body by the membrane transporter SLC6A8 (also called the Creatine Transporter: CrT).

  1. Line 41.42. "Therefore, this suggests that the BBB is semipermeable to creatine." why this statement? Please explain better.

3R: Apologies for the confusion, we removed this from the manuscript.

  1. Lines 48-50 "While the highest concentrations of creatine and the highest specific CK activity are found in skeletal muscle, the compound is responsible for far more actions in other areas of the body." Please be more specific, which are these actions? And where do they take place?

4R: In the rest of the paragraphs in this section (Lines 55-81), we specifically discussed the action and area where creatine occurs. We discussed how creatine plays a role in the nervous system, fat metabolism, and cancer. Thus, this sentence serves as a goal for opening the topic.

  1. Lines 54-56. The assertion that "creatine promotes cognitive performance [8-10]" is somehow in contrast to what is stated in lines 58-59: "Its specific role in promoting cognitive health is unclear"

5R: Our meaning here is that even though there is evidence that creatine promotes cognitive performance the actual pathway for function behind it is still a mystery. We apologize for the misunderstanding, and here are the refined sentences to make it clearer:

“Therefore, creatine transport is essential to brain health. Indeed, several studies have shown that, on a broad level, creatine promotes cognitive performance [8-10]; delay Huntington’s Disease progression [11](although this is controversial [12]), prevention of traumatic brain injury [13], and several other pathologies [14]. Despite creatine's obvious importance to CNS health, the specific pathway in promoting cognitive health still need further examination.”

  1. Line 62. What is: "it can prevent excitotoxic damage to neurons" There is no explanation for this effect. Please be clearer.

6R: Studies have been discovered supporting the idea that creatine and phosphocreatine could reduce harmful malonate and 3-nitropropionic acid, which are believed to have neurotoxicity. Below, we add this information to the sentences.

“Another mechanism suggests it can prevent excitotoxic damage to neurons from malonate lesions and 3-nitropropionic acid neurotoxicity [17].”

  1. Lines 63-68, This portion on the futile cycles in metabolism, is not contextualize.d with the title of this paper.

7R: In this section, we discuss the futile cycles to show audience another possible function of creatine. Our tile mentioned “it’s not just about energy storage”, so we think it’s better also use some work to discuss about energy storage at first before we go deeper into cancer and immunology.

  1. Lines 70-76. The authors talked about creatine in the liver. However, the English language is not good. Also, the content of the paper cited is not well explained. Indeed, this paper concludes: "The results indicate that TNFalpha-induced apoptosis was inhibited in CK transgenic mice livers by maintaining mitochondrial function" but the creatine was used in a diet of mice in a specific and artificial model compared to normal diet with less creatine.

The relevance of creatine could be on the good state of mitochondria, and this is not indicated.

8R: It’s a very good point and thanks so much for letting us know. It’s true that in the paper researchers use both wild type and CK liver-overexpression mouse to test under normal diet and creatine supplemental diet. Based on their paper we rephrase our text to add more information:

“Creatine also plays a crucial role in preventing cell death in other tissues. For instance, in tumor necrosis factor (TNF)-induced apoptosis, creatine inhibits cell death. In a study comparing hepatic injury in wild-type and liver-specific creatine kinase (CK) overexpression transgenic mice, the results demonstrated that both liver CK expression and dietary creatine supplementation effectively prevented liver cell apoptosis [20]. This discovery revealed the anti-apoptotic effects of creatine are by maintaining mitochondrial functionality under inflammatory stress[20] .”

  1. Lines 77-81. This part is not specific. The authors claimed that there are contradictory results, but they did not mention any of specific information to give at least an idea on what the authors are talking about.

9R: We are hoping to use further section to explain about what contradictory results are but yes, we agree that we should at least give reader a brief idea about it. Thus, here is the new version:

The role of creatine in tumor biology, however, is less clear. Several seemingly contradictory observations about this pathway in cancer therapy have been made. Simply put, some studies have shown that creativity is helpful in preventing cancer, while others suggest that it has a promoting effect on cancer.

Chapter 2

  1. Lines 83-135. In this part of the paper, the authors reported, in a quite confusing manner, the role of creatine as an antitumor factor. This part is a list of assertions, not well organized and presented. 

It appears that creatine can affect CD8 T cells and macrophages, besides having some direct inhibiting effect on tumor cells. Unfortunately, the authors do not make this clear distinction starting the chapter, but they insert confusing assertions. It is difficult to follow,

11R: In Chapter 2, we begin by discussing how creatine might directly prevent tumor growth. The evidence for this is minimal as there are only a few papers about it. The rest of the chapter is about how it promotes anti-tumor immunity by promoting t-cell functionality and macrophage polarization. Thus, we believe this is elaborated well in the manuscript and reflects the reality of the research. However, we edited the flow of the chapter in hopes to make these points clearer and make our assertions less strong.

  1. The statement that M1 and M2 paradigm is an artifact found in vitro is really too strong. I certainly agree that it is not possible to distinguish well between M1 and M2. However, the literature is plenty of papers claiming the difference between these two cell populations and the possibility to distinguish several M2 subsets, at least in murine models. In addition, there are several clinical trials based on this distinction trying to reconvert M2 into M1. Almost all the papers regarding the role of creatine come from the murine models (in some instances, quite difficult to reproduce and artificially built).

12R: We’ve softened the language on this.

However, it’s essential to appreciate that the M1/M2 dichotomy is not as simple as previously thought [31]. “

  1. Anyway, the flow of information is not schematic and some portions are difficult to read.

For instance, the assertion:"This fits in with the more established view of creatine in cancer; it’s a promalignant process, which we discuss in detail below" I do not consider creatine a "process". Also, it is somehow obvious that creatine (a battery for T cells) can favor tumor growth, as a tumor needs ATP. 

13R: "Thank you for pointing that out. We reworded the text to make it easier to read and more fluent. Regarding creatine’s role in promoting tumor growth, we believe it warrants careful consideration. Metabolic homeostasis often functions as a delicate balance, where deviations on either side can lead to significant effects. This is how we approach creatine uptake. There may be a competitive balance between T cells and tumors in consuming creatine, so we remain neutral on whether creatine favors or promotes cancer. Further experiments and more precise contextual analyses are needed before drawing definitive conclusions."

Chapter 3

As for the chapter 2, the chapter 3 is the description of the protumor role for creatine. 

  1. The MPS1 pathway is not described but mentioned, this reviewer can have an idea of this pathway, but all readers know any kind of biochemical pathway. The authors have to introduce the pathway and the link among the other pathways to give the right flow of information.

14R: We add more information on MPS1 to help readers understand what we are talking about:

“The supply of creatine triggers the MPS1 signaling pathway axis, which upregulates SMAD2/3 and Snail/Slug[42]. MPS1, known as Monopolar spindle 1, is a kinase participates in chromosome-spindle attachment and cell cycle checkpoints[43]. Snail and Slug play important roles in cancer metastasis by promoting epithelial‒mesenchymal transition (EMT) to improve cancer cell mobility.”

  1. Near the end of the chapter, the authors introduced the CK, but these enzymes (and the types of these enzymes) have not been considered in detail in the introduction. This makes it difficult to link the different portions of the chapter.

15R: Thanks for reminding us about this. We add some more description about CK in the introduction part and here is the added sentences:

“The traditional role of creatine is to 'store' excess phosphate through the action of creatine kinase (CK), which phosphorylates creatine to produce phosphocreatine (PCr). Four types of CK have been identified. CKMT1 and CKMT2 are mitochondrial isoforms that primarily catalyze the forward reaction, converting creatine to PCr. CK brain type (CKB) and CK muscle type (CKM) are cytoplasmic isoforms that primarily catalyze the reverse reaction, converting PCr back to creatine while generating ATP. Although these reactions are reversible, the cellular energy state often determines their direction. This system serves as a critical energy buffer, facilitating the interconversion of ADP and ATP and storing phosphate during metabolic stress [5]. “

  1. In the last chapter, the authors reported some of the data derived from relevant publications of the authors on GBM. I would suggest inserting a figure or an additional panel into figure 2. Indeed, the observations mentioned are of relevance to GBM biology.

The font in the figures is small and difficult to see.

16R:

Thanks for the suggestion. We re-structured our graph to make it easier to read and created a new figure to represent our finding of creatine & GBM.

Reviewer 2 Report

Comments and Suggestions for Authors

This work is extremely important, since in recent years, we have lived in the era of food supplementation.  Especially, on social media, creatine supplementation is exalted, without professional monitoring, being described as without health risks.

1- Even though it is a narrative review, I think it is important to describe how this literature review was carried out.

"For example: Methodology

Suitable studies were found through the use of the electronic search systems PubMed, Google Scholar, and Scopus for the literature search. We also searched the bibliographies to identify relevant studies and reviews. The database search was conducted by combining search terms “creatine” and “creatine” or “CNS” or “phosphocreatine” in combination with “cancer”, “Tumor metabolism”, “Brain tumor” paired with  “Medicinal use”, “Pharmacokinetics” and “Toxicity profile”. The articles were assessed based on pharmacological research on creatine and in vitro and in vivo experiments to determine the best search. Additionally, research and review articles were excluded for being outside the scope of this article.

2-I suggest some new articles on the subject that can add some information to this review article that they have been published in 2024.

Ushigome M, Shimada H, Kaneko T, Miura Y, Yoshida K, Suzuki T, Kagami S, Kurihara A, Funahashi K. Preoperative Low Creatine Kinase as a Poor Prognostic Factor in Patients with Colorectal Cancer. J Gastrointest Cancer. 2024 Sep;55(3):1212-1219. doi: 10.1007/s12029-024-01069-9. Epub 2024 Jun 13. PMID: 38869820.

Chen L, Qi Q, Jiang X, Wu J, Li Y, Liu Z, Cai Y, Ran H, Zhang S, Zhang C, Wu H, Cao S, Mi L, Xiao D, Huang H, Jiang S, Wu J, Li B, Xie J, Qi J, Li F, Liang P, Han Q, Wu M, Zhou W, Wang C, Zhang W, Jiang X, Zhang K, Li H, Zhang X, Li A, Zhou T, Man J. Phosphocreatine Promotes Epigenetic Reprogramming to Facilitate Glioblastoma Growth Through Stabilizing BRD2. Cancer Discov. 2024 Aug 2;14(8):1547-1565. doi: 10.1158/2159-8290.CD-23-1348. PMID: 38563585.

He L, Lin J, Lu S, Li H, Chen J, Wu X, Yan Q, Liu H, Li H, Shi Y. CKB Promotes Mitochondrial ATP Production by Suppressing Permeability Transition Pore. Adv Sci (Weinh). 2024 Aug;11(31):e2403093. doi: 10.1002/advs.202403093. Epub 2024 Jun 19. PMID: 38896801; PMCID: PMC11336976.

3- in vitro and in vivo must appear in italics in the text.

Author Response

Reviewer #2

This work is extremely important, since in recent years, we have lived in the era of food supplementation.  Especially, on social media, creatine supplementation is exalted, without professional monitoring, being described as without health risks.

1- Even though it is a narrative review, I think it is important to describe how this literature review was carried out.

"For example: Methodology

Suitable studies were found through the use of the electronic search systems PubMed, Google Scholar, and Scopus for the literature search. We also searched the bibliographies to identify relevant studies and reviews. The database search was conducted by combining search terms “creatine” and “creatine” or “CNS” or “phosphocreatine” in combination with “cancer”, “Tumor metabolism”, “Brain tumor” paired with  “Medicinal use”, “Pharmacokinetics” and “Toxicity profile”. The articles were assessed based on pharmacological research on creatine and in vitro and in vivo experiments to determine the best search. Additionally, research and review articles were excluded for being outside the scope of this article.

1R: Thanks so much for the advice! We have included a methodology section to the manuscript, below is what was added:

To explore the role of creatine in glioblastoma (GBM), we reviewed our own data and noted the lack of a comprehensive resource for new researchers in this field. We first compiled relevant papers we had read and developed a preliminary structure for the manuscript, including an introduction, sections on the positive and negative roles of creatine in cancer, and a discussion. Next, we conducted literature searches on PubMed and Google Scholar using keywords such as “creatine,” “phosphocreatine,” and “creatine kinase,” in combination with terms like “cancer” and “T-cell immunology.” Review articles were examined to gain a broader understanding of creatine and identify knowledge gaps. From there, we focused on original research articles by following citations from review papers and targeted searches. This approach allowed us to build a comprehensive foundation for our manuscript.

2-I suggest some new articles on the subject that can add some information to this review article that they have been published in 2024.

Ushigome M, Shimada H, Kaneko T, Miura Y, Yoshida K, Suzuki T, Kagami S, Kurihara A, Funahashi K. Preoperative Low Creatine Kinase as a Poor Prognostic Factor in Patients with Colorectal Cancer. J Gastrointest Cancer. 2024 Sep;55(3):1212-1219. doi: 10.1007/s12029-024-01069-9. Epub 2024 Jun 13. PMID: 38869820.

Chen L, Qi Q, Jiang X, Wu J, Li Y, Liu Z, Cai Y, Ran H, Zhang S, Zhang C, Wu H, Cao S, Mi L, Xiao D, Huang H, Jiang S, Wu J, Li B, Xie J, Qi J, Li F, Liang P, Han Q, Wu M, Zhou W, Wang C, Zhang W, Jiang X, Zhang K, Li H, Zhang X, Li A, Zhou T, Man J. Phosphocreatine Promotes Epigenetic Reprogramming to Facilitate Glioblastoma Growth Through Stabilizing BRD2. Cancer Discov. 2024 Aug 2;14(8):1547-1565. doi: 10.1158/2159-8290.CD-23-1348. PMID: 38563585.

He L, Lin J, Lu S, Li H, Chen J, Wu X, Yan Q, Liu H, Li H, Shi Y. CKB Promotes Mitochondrial ATP Production by Suppressing Permeability Transition Pore. Adv Sci (Weinh). 2024 Aug;11(31):e2403093. doi: 10.1002/advs.202403093. Epub 2024 Jun 19. PMID: 38896801; PMCID: PMC11336976.

2R: Thanks so much for the advice and it really help us a lot. After carefully go through those references, we added them into our new paper.

3- in vitro and in vivo must appear in italics in the text.

3R: We adjusted those formatting issue and thanks very much

Reviewer 3 Report

Comments and Suggestions for Authors

The review article was generally written well.  The review article addresses an important issue for creatine in cancers. Since creatine and its product phosphocreatine are essential for muscle function, creatine role in cancer development and progression is necessary to explore and future discussion. The reference is listed appropriately.

Comments:

1. Figure 2: Any formation for creatine and its pathway in pancreatic and colon cancers.

2. Any information for creatine and its pathway affects p53 and p53/MDM2 pathway?

3. Studies have shown that most cancers use anaerobic glycolysis to produce ATP. Does creatine affect anaerobic glycolysis in cancers?

Author Response

Reviewer #3

The review article was generally written well.  The review article addresses an important issue for creatine in cancers. Since creatine and its product phosphocreatine are essential for muscle function, creatine role in cancer development and progression is necessary to explore and future discussion. The reference is listed appropriately.

Comments:

  1. Figure 2: Any formation for creatine and its pathway in pancreatic and colon cancers.

1R: The role of creatine in colon cancer is like that in liver cancer, both promoting cancer metastasis by providing phosphorylated energy. An interesting point in pancreatic cancer is that the involvement of SLC6A8 will damage the dendritic cell formation, which leading to the impairment of the immune function against cancer. We have included more relevant text and references about these studies.

  1. Any information for creatine and its pathway affects p53 and p53/MDM2 pathway?

2R: That’s a very good point; there’s a study about GAMT and p53, which we have included in the manuscript. Also, studies in Hela cells and cancer cells have shown that p53 overexpression could inhibit CKB promoter and may play a role in tumorigenesis. That information is also included in the manuscript.

  1. Studies have shown that most cancers use anaerobic glycolysis to produce ATP. Does creatine affect anaerobic glycolysis in cancers?

3R: Based on our own work, and others, this answer is not yet known. As mitochondrial CKMT and the creatine phosphagen system are so intricately linked, this has been the previous focus of most work. Furthermore, while the Warburg effect is real (tumor cells produce lactate even in the presence of oxygen), both glycolysis and mitochondrial function are essential for tumor growth (PMID: 32694689). However, Considering the intricate connection between these metabolic pathways, it would not surprise me if creatine metabolism was connected to glycolysis.

Round 2

Reviewer 1 Report

Comments and Suggestions for Authors

The Authors improved the manuscript by replying to reviewers' suggestions.

Reviewer 3 Report

Comments and Suggestions for Authors

The authors answered my questions. No more comments.